# Risk Factors Associated with Pancreatic Cancer in the UK Biobank Cohort

**DOI:** 10.3390/cancers14204991

**Published:** 2022-10-12

**Authors:** Te-Min Ke, Artitaya Lophatananon, Kenneth R. Muir

**Affiliations:** Division of Population Health, Health Services Research and Primary Care, School of Health Sciences, Faculty of Biology, Medicine and Health, The University of Manchester, Manchester M13 9PT, UK

**Keywords:** pancreatic cancer, risk factors, modifiable risk factors, non-modifiable risk factors, UK Biobank cohort

## Abstract

**Simple Summary:**

The study explored pancreatic cancer (PaCa) risk factors in the UK Biobank cohort. The risk factors included non-modifiable risk factors: age, gender, and modifiable risk factors: cigarette smoking, overweight and obesity, increased waist circumstance, abdominal obesity, Diabetic Mellitus (DM), and pancreatitis history. The study findings suggested that stopping cigarette smoking, avoiding overweight or obesity, abdominal obesity, Diabetic Mellitus, and pancreatitis history could contribute to a significant reduction in future PaCa cases if these exposures are avoided.

**Abstract:**

Evidence on pancreatic cancer (PaCa) risk factors from large population-based cohort studies is limited. This study investigated the PaCa risk factors and the population attributable fraction (PAF) of modifiable risk factors in the UK Biobank cohort. The UK Biobank is a prospective cohort consisting of 502,413 participants with a mean follow-up time of 8.2 years. A binomial generalized linear regression model was used to calculate relative risks for PaCa risk factors. PAF was calculated to estimate the proportional reduction in PaCa if modifiable risk factors were to be eliminated. A total of 728 (0.14%) PaCa incident cases and 412,922 (82.19%) non-PaCa controls were analyzed. The non-modifiable risk factors included age and gender. The modifiable risk factors were cigarette smoking, overweight and obesity, increased waist circumstance, abdominal obesity, Diabetic Mellitus (DM), and pancreatitis history. The PAF suggested that eliminating smoking and obesity can contribute around a 16% reduction in PaCa cases while avoiding abdominal obesity can eliminate PaCa cases by 22%. Preventing pancreatitis and DM could potentially reduce PaCa cases by 1% and 6%, respectively. This study has identified modifiable and non-modifiable PaCa risk factors in the UK population. The PAF of modifiable risk factors can be applied to inform PaCa prevention programs.

## 1. Introduction

Pancreatic cancer (PaCa) is the 12th leading cancer [1] and the 7th major cause [1,2] of cancer death globally. According to 2020 Global Cancer Statistics [2], the incidence rate of PaCa in high/very high human development index (HDI) countries is 4–5 fold higher than in low/medium HDI countries. In the United Kingdom, PaCa is the 10th most common cancer, with approximately 10,452 people diagnosed annually (year 2016–2018) [3]. With the advancement of cancer treatment and generalized screening, the survival rates for common cancers such as breast and colorectal cancer [4] are improving. However, there has not been significant progress in the survival rate of PaCa. The 1-year survival rate of PaCa is approximately 28%, and the 5-year survival rate is only about 6% [5].

Previously published literature [6,7,8] has reported many potential PaCa risk factors. Some non-modifiable risk factors showed probable evidence of PaCa, including ageing, male gender, African American ethnicity, non-O blood type, family history, inherited syndromes including Peutz-Jeghers syndrome, Hereditary pancreatitis and Lynch syndrome, and germline mutation (CDKN2A, TP53, MLH1, ATM, BRCA2, MSH2, MSH6, PALB2, and BRCA1). Other modifiable risk factors with probable or convincing evidence are cigarette smoking, heavy alcohol consumption, increased Body Mass Index (BMI) and abdominal obesity [9,10], chronic pancreatitis, Diabetic Mellitus (DM), hepatitis B, cholecystectomy [11,12], and periodontal disease. Nevertheless, there were still other ambiguous risk factors with inconclusive evidence, such as increased consumption of processed meat, Helicobacter Pylori infection, and Systemic Lupus Erythematosus (SLE) history [13,14]. It has thus been suggested that 37% of pancreatic cancer patients in the United Kingdom can be prevented [15].

To date, there is a lack of evidence from large population-based cohort studies. Moreover, evidence on the population attributable fraction (PAF) among the general population for these factors is limited. PAF is defined as the proportion of specific disease incidence that can be attributed to a particular exposure.

In this study, we aimed to explore potential risk factors, including modifiable and non-modifiable risk factors of PaCa in the UK population. In particular, we also emphasized the PAF of the modifiable risk factors in the population, which could contribute to increasing the public health awareness messages.

## 2. Materials and Methods

### 2.1. Study Population and Study Design

UK Biobank is a UK national-based dataset containing lifestyle, genetic, and various health information, which recruited around half a million participants from the community from the year 2006 to 2010. The age of all participants when first attending the assessment center ranged from 37 to 73, and the clinical outcome follow-up data have continued to date. The UK Biobank aims to be used to improve human health by identifying enhancements in medicine, treatment, and scientific evidence on common diseases [16]. More details on the UK Biobank can be found at http://www.ukbiobank.ac.uk/ (accessed on 22 August 2022). The UK Biobank cohort data enrolled 502,413 participants with a mean follow of 8.2 years (till 31 March 2017).

### 2.2. Defining Pancreatic Cancer Cases and Non-Pancreatic Cancer Controls

Pancreatic cancer was defined as a malignant neoplasm of the pancreas. The International Classification of Diseases 9 and 10 (ICD9 and ICD10) code and self-reported data were used to record all the subtypes and different anatomic parts of pancreatic malignancies. The occurrence of pancreatic cancer was recorded with the time of diagnosis. The details of codes to identify pancreatic cancer cases are summarized in Appendix A.

#### 2.2.1. Pancreatic Cancer Cases

All pancreatic cancer cases were identified by combing all three data sources, including ICD9,10 and self-reported data. As this is a cohort study, follow-up data were available for each source. There were 11 follow-up time points for the ICD9, 14 follow-up time points for the ICD10 code, and 14 follow-up time points for the self-reported data. The codes related to pancreatic cancer are listed in Appendix A. The incident and prevalent cases were distinguished by comparing ‘the age when participants attended the study’ with ‘the age when they were first reported pancreatic cancer’. Prevalent cases were defined while ‘the participant’s age of attending’ was greater than ‘the age of pancreatic cancer diagnosis’. Incident cases were identified if ‘the participant’s age of attending’ was less than ‘the age of pancreatic cancer diagnosis’. As for the self-reported data, the prevalent cases were defined as ‘the interpolated age of the participant when cancer was first diagnosed’ was smaller than ‘the participant’s age of attending the assessment center’. Conversely, the incident cases were considered if ‘the interpolated age of the participant when cancer was first diagnosed’ was greater than the participant’s age of attending the assessment center’.

First, case status (incident or prevalent) was determined from each source. Next, we applied the following criteria:If the participant was an incident case in at least one of three sources and was not defined as a prevalent case in any of 3 data sources, this case was classified as an incident case.If the participant appeared as the prevalent case in at least one of three different sources and was not defined as an incident case in any of 3 data sources, then this case was categorized as a prevalent case.

In sum, of 819 pancreatic cases, 728 cases were identified as incident cases and 91 cases as prevalent cases. In this study, only PaCa incident cases were included in the analysis.

#### 2.2.2. Non-Pancreatic Cancer Controls

The participants with no records of neoplasms, in situ neoplasms, benign neoplasms, and neoplasms of uncertain or unknown behavior were classified as our non-pancreatic cancer controls (412,922).

#### 2.2.3. Exclusion Criteria

In the pancreatic cancer case group, 91 prevalent cases were excluded. In the non-pancreatic cancer control group, 88,672 participants (have any records of other neoplasms, in situ neoplasms, benign neoplasms and neoplasms of uncertain or unknown behavior) were excluded.

### 2.3. Exposures

The exposures were categorized into modifiable and non-modifiable factors. Modifiable factors were defined as the exposures that occurred before PaCa diagnosis and could be preventable or modifiable. Non-modifiable factors were identified as the exposure factors that were presented before the PaCa diagnosis and could not be prevented or modified. Non-modifiable variables were gender, age, and ethnic group. Modifiable variables included lifestyle-related variables: cigarette smoking status, alcohol intake frequency, processed meat consumption frequency, Body Mass Index (BMI), waist circumstance and Waist-to-Hip ratio (WHR), and medical history-related variables: pancreatitis, Diabetes Mellitus (DM), hepatitis B, cholecystitis, Helicobacter Pylori (H. pylori) infection, and Systemic Lupus Erythematosus (SLE).

### 2.4. Statistical Analysis

A binomial generalized linear regression model was employed to compute relative risks (RR) and 95% confidence intervals (95% CI). The pancreatic cancer variable as a dependent variable was coded as a binary variable (pancreatic cancer cases and non-pancreatic cancer controls). Each independent variable was described and classified as shown in Appendix A. The analyses were adjusted for age and gender. Student’s *t*-test was used to compare mean values between the case and control groups. Chi-square test was used to explore the difference between observed and expected of categorical data. *p*-value less than 0.05 was considered as “statistically significant”. A 95% CI not including 1 was also used to guide delineation of statistical significance.

To calculate the PaCa incident rate in the UK Biobank cohort, the STATA command (stptime) was used to derive the overall person-time of observation and PaCa incident rate. The endpoint time for each participant was defined as either the date of PaCa diagnosis or the last follow-up date on 31 March 2017. Furthermore, the population attributable fraction (PAF) was calculated to evaluate how many cases could be prevented by eliminating the significant modifiable risk factors related to PaCa. PAF was calculated for only the significant modifiable risk factors among the whole cohort. PAF was also calculated for the subgroup of the population which was considered as “exposed”. The STATA command (punaf) was used [17].

STATA version 17 for Windows was used to perform all the statistical analyses [18]. Results with *p*-value < 0.05 and 95% confident intervals (95% CI) not including one were considered statistical significance in this study.

## 3. Results

There was a total of 502,413 participants (229,085 male and 273,328 female participants). The mean age of participants when entering the cohort was 56.53 years (SD ±8.10). The mean follows up time up to 31 March 2017 was 8.18 years (SD ± 0.86). The total number of PaCa incident cases was 728 (0.14%), with 388 males and 340 females (Table 1). 412,922 non-PaCa controls accounted for 82.19%, which consisted of 218,357 females and 194,565 males (Table 1). A total of 88,672 (17.65%) participants with other neoplasms and 91 (0.02%) PaCa prevalent cases were excluded (Appendix A). The PaCa incidence rate of the whole cohort was 0.18 per 1000 person-years.

### 3.1. Demographic Characteristic Distributions

The distributions of the demographic characteristic were compared between the PaCa cases and non-PaCa controls in Table 1. In the PaCa cases, there was a slightly higher proportion of males (53.30%) than females (46.70%). However, in the controls, there were more females (52.88%) than males (47.12%). The mean age of cases was approximately 66 years old, which was significantly older than the controls (~64 years of age) (Student’s *t*-test *p*-values < 0.05). For ethnicity, the white population accounted for the vast majority of cases (96.41%) and controls (93.97%). For the modifiable factors, the proportion of former smokers, current smokers, and participants who consumed alcohol daily were all higher in the cases than in the controls (*p*-value < 0.05). On average, both former and current smokers smoked more cigarettes per day in the cases compared to the controls (Student’s *t*-test *p*-value < 0.05). The proportion of overweight, obese, and abdominally obese individuals were greater in the cases compared to the controls. (*p*-value < 0.05). The mean values of waist circumference and WHR were greater in the cases as compared with the controls (Student’s *t*-test *p*-values < 0.05). Regarding the processed meat consumption frequency, there was no significantly different distribution between PaCa cases and non-PaCa controls. For medical history-related variables, the percentage of participants with pancreatitis or DM was higher in the PaCa cases than in the non-PaCa controls (*p*-value < 0.05). On the other hand, there were no significant differences in the distribution of hepatitis B, cholecystitis, H. pylori infection, and SLE between PaCa cases and non-PaCa controls.

### 3.2. Relative Risks of the Non-Modifiable Factors

Relative risks (RRs) of the non-modifiable factors are shown in Table 2. For gender, men were at increased risk of PaCa by 27% (RR = 1.27, 95% CI: 1.10–1.47) compared to women after adjusting for age. For the age variable, the result showed a 3% higher risk of developing PaCa with increasing one year of age (RR = 1.03, 95% CI: 1.02–1.04). Breakdown by the ethnic group did not show any association with PaCa risk (all ethnic groups 95% CI included 1).

### 3.3. Relative Risks of the Modifiable Lifestyle-Related Factors

Table 2 and Table 3 show the results of the modifiable risk factors. Former cigarette smokers were at a 36% increase in PaCa risk (RR = 1.36, 95% CI: 1.13–1.62). Current cigarette smokers had about a two-fold greater risk of PaCa (RR = 2.16, 95% CI: 1.71–2.73) compared to never-smokers (Table 2). Moreover, smoking one additional cigarette per day was associated with an increased PaCa risk of 2% for former smokers and of 4% for current smokers (*p*-value < 0.0001) (Table 3). Furthermore, the results of the daily cigarette smoking numbers, which were stratified into five different groups (Table 3), showed that former smokers who smoked 16–20 cigarettes per day increased PaCa risk by 40% (RR = 1.40, 95% CI: 1.08–1.82), and those who smoked greater than or equal to 21 cigarettes/per day raised two times PaCa risks (RR = 2.03, 95% CI: 1.54–2.68), comparing to never smokers. Among the current smokers (Table 3), those who smoked 11–15, 16–20, and ≥21 cigarettes per day increased risk of PaCa (RR = 2.34, 95% CI: 1.54–3.58; RR = 2.61, 95% CI: 1.77–3.87; RR = 3.46, 95% CI: 2.24–5.35) compared to non-smoker participants, respectively.

Regarding the association between BMI and PaCa risk, overweight and obese participants were at increased PaCa risks with RR = 1.23 (95% CI: 1.03–1.48) and RR = 1.39 (95% CI: 1.14–1.69) compared to people whose BMI less than 25, respectively. For waist circumference, one centimeter increment of waist circumstance showed a 2% increase in PaCa risk (RR = 1.02, 95% CI: 1.01–1.02). Abdominal obesity participants increased PaCa risk by 56% (RR = 1.56, 95% CI: 1.32–1.84) compared to participants with normal WHR. There was no significant relationship between alcohol intake frequency, processed meat consumption and the risk of PaCa in this study.

### 3.4. Relative Risks of the Modifiable Medical History-Related Factors

Results are shown in Table 2. Participants who had a medical history of pancreatitis had an approximate six times increased PaCa risk (RR = 5.54, 95% CI: 2.88–10.65) compared to participants without pancreatitis. Additionally, the medical history of DM was significantly associated with a higher risk of PaCa (RR = 2.08, 95% CI: 1.64–2.63). Medical history of cholecystitis, H. pylori, hepatitis B, and SLE were not associated with the PaCa risk (all 95% CI included 1).

### 3.5. Population Attributable Fraction (PAF) of Modifiable Risk Factors

Results of PAF for significant modifiable risk factors in the whole population and sub-population are shown in Table 4. For the lifestyle-related risk factors, smoking cessation could eliminate PaCa up to 16% in the general population and up to 54% in the current smoker group. Maintaining BMI to less than 30, could prevent PaCa up to 17% in the general population and 28% in the obese population. In addition, 22% and 36% of PaCa could be prevented by avoiding abdominal obesity (criteria of WHR ≥ 0.90 in males and ≥0.85 in females) in the general population and abdominal obesity population, respectively. For the medical history-related factors, preventing pancreatitis could reduce 1% and 82% of PaCa cases among the general population and people with pancreatitis, respectively. 6% and 52% of PaCa could be eliminated by avoiding DM in the general population and people with a DM history, respectively.

## 4. Discussion

This study explored PaCa risk factors in the UK Biobank cohort. The relative risks and PAF are reported for each risk factor. The analysis was exploratory.

In the study cohort, males had a higher PaCa risk, which has also been reported in the CRUK statistics [15,19]. In addition, this pattern was similar to the 2018 Global Cancer Statistics [1]; the age-adjusted PaCa incidence rate in males (5.5 per 100,000 men) was higher than in females (4.0 per 100,000 women). Modifiable lifestyle-related, environmental, and occupational factors might be considered to contribute to the higher PaCa risk in men [20,21]. PaCa risk was observed to rise with age in this study, and the median age of diagnosis of PaCa was 67 years old. The incident rate of PaCa has been previously reported to rise with age [15,19,20,21,22]. The ageing process is generally thought to lead to DNA damage; however, it is also often caused by exposure to lifestyle-related risk factors [23,24].

Some descriptive statistics [25,26] have displayed that African Americans had a significantly higher age-adjusted PaCa incidence rate. In this study, the Caucasian population accounted for the majority (94%) of this cohort. Hence, the investigation of the PaCa incidence rate disparities among different races still needs further investigation in data sets with larger numbers in the different ethnic groups. Other familial-related factors such as non-O blood type, a kindred PaCa history, several familiar syndromes, and germline mutation have been reported to be associated with higher PaCa risk [6,7,8]. However, some of the information is not available to obtain from the UK Biobank. Furthermore, the focus of this paper was on epidemiological factors for pancreatic cancer.

### 4.1. Modifiable Lifestyle-Related Risk Factors

The dominant relationship between cigarette smoking and higher PaCa risk was evident in this study. Current cigarette smokers and ex-smokers had an increased risk of PaCa. Many previous studies [27,28,29] have also demonstrated a relationship between tobacco smoking and PaCa risk. Tobacco degradation products have been well known to participate in the signaling cascade of angiogenesis, tumor cell growth, and tumor metastasis [30,31]. In addition, tobacco cigarette smoking was reported [32] to increase the mutational burden for numerous mutational traits, which may be associated with carcinogenesis.

Being overweight or obese significantly increased PaCa risk. Overweight and obese participants showed a 23% and an approximate 40% increased risk of PaCa, respectively. A meta-analysis study [33] also concluded that people with a per 5 kg/m^2^ BMI increase was related to a higher PaCa risk (RR = 1.12; 95% CI: 1.06–1.17). Moreover, abdominal obesity (as measured by waist circumference or waist-to-hip ratio) has also emerged as an important risk factor for PaCa in recent years [9,10]. In this study, an increment of one centimeter of waist circumstance was found to elevate the risk of PaCa. Moreover, abdominal obesity participants had a 1.6 times increased risk of PaCa compared to the participants with normal WHR. A similar conclusion that either increasing waist circumference or WHR was associated with a higher risk of PaCa has been previously reported in a systematic review [9].

The pathogenetic mechanism of elevating PaCa risk in obese individuals is likely complicated. At the biological level, obesity is known to be involved [34] in carcinogenesis and inflammation by releasing the potential pro-carcinogenic mediators such as adipokines, vascular endothelial growth factor (VEGF), and insulin-like growth factor (IGF). Obesity is often caused by potentially modifiable lifestyle-associated factors such as low activity, poor lifestyle, and unhealthy diet habits. These are strongly associated with metabolic syndrome, which is known [35,36] to elevate PaCa risk.

In this study, the PAF is presented for the whole cohort, and for the exposed subgroup population. Our findings of the former demonstrated a PAF of 16% for current tobacco smoking and 10% for former smokers in PaCa. A previous UK study [21] reported a PAF of 21.9% for tobacco smoking in PaCa, and another Korean study [37] reported a PAF of 15.5%. The PAF from these previous studies was however estimated through the published summary data of RRs from literature reviews. The discrepancy seen in these studies compared to ours could also be due to different disease incidence rates and exposure prevalent rates.

Moreover, we separately showed the PAF of tobacco smoking by concentrating on the current and former smoker population. Our results suggest that 50% of PaCa cases in current smokers and 25% of PaCa cases in former smokers could be prevented if they had never smoked. Regarding obesity, our results showed PAFs of 11%, 17%, and 22% for overweight, obesity, and central obesity, respectively. The previous UK study [21] demonstrated a PAF of 12.3% for overweight and obesity in PaCa, and another review study [38] reported a median PAF of 8%. In that review study, the authors used various sources to obtain imputed RR, incidence of disease, and prevalence of exposure. To our knowledge, there is no study presenting PAF separately for a population of overweight, obesity, and central obesity. In our exposed sub-population analyses, our findings suggested that obese people could potentially decrease around 30% of the PaCa cases by reducing their BMI to lower than 30, and overweight people could avoid about one-fifth of PaCa cases if they maintained a normal BMI. Additionally, abdominal obesity males and females could potentially avoid more than one-third of PaCa cases by maintaining WHR less than 0.9 (males) and 0.85 (females), respectively.

The frequency of alcohol intake and processed meat consumption did not show any association with the risk of PaCa in our study. Several studies [27,39] have shown that an increased PaCa risk is related to high (≥24 g/day) and heavy (>39 g/day) alcohol consumption. However, our findings could be due to the limitation on data of the amount of alcohol drinking in this study. Data related to alcohol consumption was available only in the frequency format and not the consumption quantity. The evidence on the relationship between PaCa risk and processed meat consumption was limited in the previous literature; therefore, further studies are needed to explore this factor.

There have been two previous large population-based studies that investigated lifestyle factors; however, their approaches to classified exposures were different to our study. The European Prospective Investigation into Cancer (EPIC) [40] cohort study recruited participants across Europe from 1992 to 2000 and followed up for 15 years. The EPIC study applied a healthier lifestyle habit (higher Healthy Lifestyle Index (HLI)). The HLI is a score calculated by combining different lifestyle habits, including smoking, adiposity, dietary pattern, alcohol intake, and physical activity. The findings suggested that participants with a higher score had lower hazard ratios (HR) of PaCa. The other study was the US Women’s Health Initiative (WHI) cohort study [41]. The study recruited postmenopausal women between 1993 and 1998 and followed up until 2020. The authors reported that higher adherence to healthier lifestyle habits (higher HLI scores) was associated with a lower risk of PaCa. In our study, we, however, explored each lifestyle-related risk factor independently. Our findings suggested that former cigarette smoking, current cigarette smoking, overweight, obesity, and central obesity were significantly associated with higher PaCa risk, respectively.

### 4.2. Modifiable Medical History-Related Risk Factors

Pancreatitis and DM history were significant risk factors in this study. Participants with a pancreatitis history had a six-fold higher risk of PaCa than those with no pancreatitis history participants. These findings are in keeping with the previous literature. A pooled analysis [42] showed that both people with pancreatic history within two years and beyond two years after diagnosis had higher OR (13.6 and 2.7) of PaCa. Likewise, another systemic review [43] also concluded that patients with pancreatitis were associated with higher PaCa risk, which is considerably elevated within two years after diagnosis of pancreatitis. Our results of a pancreatitis history related to higher PaCa risk are consistent with previous studies [42,43]. Either acute or chronic pancreatitis is led by the over-activated digestive enzymes that damage the pancreatic cells and which might participate in the progression of chronic inflammation, tumor carcinogenesis and mutation [44,45]. In an experimental study in mice, the expression of oncogenetic KRAS was found [46] to be related to acute pancreatitis, which may develop into PaCa. Additionally, our PAF result indicated that avoiding pancreatitis could eliminate 1% and at least 80% cases of PaCa in the general population and in pancreatitis patients, respectively. In this study, we observed a wide confident interval and the discrepancy of PAF values between the general population and people with pancreatitis, which might be due to a small number of pancreatitis in the cohort.

In this study, the results demonstrated a two times risk of PaCa among participants with DM history. A meta-analysis [47] demonstrated that long-term (≥5 years) DM patients raised around a 1.6 times PaCa risk, and another pancreatic cancer cohort consortium (PanScan) [48] pooled analysis concluded that PaCa risk was elevated with an OR of 1.8 among patients with 2 to 8 years DM history. In addition, a previous cohort study [49] also revealed that new-onset DM cases also had a higher RR of 2.2 PaCa risk. Although distinguishing the new-onset, long-term, type 1, type 2, and type 3c DM status is not possible in this cohort, the conclusion that DM history is associated with an elevating PaCa risk is still consistent with other studies [47,48,49]. Furthermore, preventing DM could reduce 6% and at least 50% of PaCa cases in the general population and in people with DM. Our results suggested that preventing DM could avoid more PaCa incidence cases. In terms of anti-diabetic medications, although there is some evidence showing that these medications are linked to the PaCa risk, the evidence is inconclusive. For example, some studies have supported that taking insulin and sulfonylureas (SUs) may increase PaCa risk [50,51]. In contrast, metformin has been hypothesized to have a protective effect on PaCa risk [52,53]. A systematic review [54] showed no relationship between using metformin (OR = 0.76, 95% CI: 0.57–1.03), insulin (OR = 1.59, 95% CI: 0.85–2.96), and thiazolidinediones (TZDs) (OR = 1.02, 95% CI: 0.81–1.30), and risk of developing PaCa. On the other hand, patients using SUs had a higher OR with 1.70 (95% CI: 1.27–2.28). In contrast, a meta-analysis study [52] did not find a significant relationship between using SUs and PaCa risk. Furthermore, the interpretation of these results needs to consider reverse causality and protopathic bias [55]. Therefore, it still needs more prospective observational studies to evaluate the impact of anti-diabetic medications on PaCa risk. In our study, we did not investigate the effects of these anti-diabetic medications due to data availability.

Other medical histories, including cholecystitis, hepatitis B, H. pylori infection, and SLE, did not show any association with the development of PaCa in this cohort study. Previous studies [11,12] have reported that patients who undertook cholecystectomy with a history of cholecystitis had a higher PaCa risk; however, this is limited to obtaining information on the surgical history of cholecystectomy in this study. Two meta-analyses studies [56,57] have demonstrated that people with hepatitis B surface antigen (HBsAg) positivity or HBV DNA positivity were correlated to a higher risk of PaCa. Nevertheless, the status of HBsAg and HBV DNA were not available in the UK Biobank. Furthermore, equivocal conclusions on the association between H. pylori infection, SLE, and the risk of PaCa have been presented in previously published meta-analysis studies (case-control and cohort studies) [13,14,58,59]. Therefore, additional high-quality and long-term follow-up studies are required to explore these exposures. For other potential medical-related risk factors, periodontal disease was associated with a higher risk of PaCa [6,7], which may be related to the alterations of the oral microbiome; however, the mechanism has not been well established. Nevertheless, we are limited to obtaining this information in the UK Biobank.

### 4.3. Strengths and Limitations

This study has several strengths. First, we used the UK Biobank study, which is a large UK-wide cohort study with a follow-up time of 8 years. The UK Biobank has a low non-response rate and low loss to follow-up. Furthermore, the result explores the modifiable risk factors, including lifestyle-related and medical history-related factors, that could provide important information to clinicians, researchers, and policymakers, and aid prevention strategies, public education, and further risk prediction model development. Ultimately, the calculation of PAF of modifiable risk factors among the general population and specific sub-population is also novel in the UK Biobank cohort. The PAF estimation can provide a more straightforward way to interpret the relative risk of certain modifiable risk factors into how many cases could be prevented in the future by adjusting specific lifestyle habits. As a consequence, this information could be used in translating scientific evidence into PaCa preventive actions.

On the other hand, there are some limitations to this study. First, a ‘healthy volunteers’ selection bias in the UK Biobank cohort is reported [60], indicating that the results may not fully represent the whole UK population, although the PaCa incidence rate was similar between the CRUK statistics [3] (17 per 100,000 people, 2016–2018) and this UK Biobank cohort (0.18 per 1000 person-year). Healthy volunteer selection bias, however, could not be ruled out. This could have an impact on underestimated risks. Second, some potential confounders may still exist in this study, which should be considered in the future PaCa risk predictive model establishment. Nonetheless, this study intended to explore the modifiable risk factors that could be applied in future prevention approaches. Finally, the non-cancer illness data were obtained by the participants’ self-report information alone. Nevertheless, compared with other UK Biobank cohort study projects, the non-cancer illness reported from ICD9 and ICD10 has not added many cases to the self-report data. For example, DM is considered a chronic disease; some may be managed by primary care, not secondary care (with ICD 9 and 10 code). Therefore, self-reported data can compensate for these under-reported cases. Finally, most of the participants were enrolled from the Caucasian population. Therefore, our conclusion should be extrapolated with caution to non-Caucasian populations.

## 5. Conclusions

In conclusion, we investigated risk factors related to pancreatic cancer in the UK Biobank cohort study. The key risk factors for PaCa can be categorized into non-modifiable risk factors, including age and gender, and modifiable risk factors contained lifestyle-related factors: cigarette smoking, overweight and obesity BMI, increased waist circumstance, and abdominal obesity with elevated WHR, and medical history-related factors: DM and pancreatitis history. Furthermore, the PAF of modifiable risk factors can also be translated into public PaCa prevention programs on how many PaCa cases can be prevented. With an increasing incidence of PaCa and still poor survival rate globally, there is an imperative need to identify approaches to prevent pancreatic cancer. Despite some risk factors being inherited or non-modifiable, there are still many modifiable risk factors, either lifestyle-related or medical history-related. This study provides further evidence to healthcare professionals, clinicians, and policymakers. These findings can be used to increase public awareness of PaCa risk factors, identify at-risk populations, and assist in early prevention approaches.

## Figures and Tables

**Table 1 cancers-14-04991-t001:** Demographic characteristics of pancreatic cancer (PaCa) case group and non-pancreatic cancer (Non-PaCa) control group.

Characteristic Variables	No. (PaCa Cases/Non-PaCa Controls)	Pancreatic Cancer Cases	Non-Pancreatic Cancer Controls	*p*-Value *
Gender	(728/412,922)			0.001
Female		340 (46.70%)	218,357 (52.88%)	
Male		388 (53.30%)	194,565(47.12%)	
Age ^#^	(728/412,922)	66.04	64.07	<0.0001
Ethnic group	(725/410,536)			0.117
White		699 (96.41%)	385,766 (93.97%)	
Mix		3 (0.41%)	2610 (0.64%)	
Asian		10 (1.38%)	9223 (2.24%)	
Black		6 (0.83%)	7406 (1.80%)	
Chinese		3 (0.41%)	1429 (0.35%)	
Other		4 (0.55%)	4102 (1%)	
Cigarette Smoking status	(613/349,478)			<0.0001
Never		330 (53.83%)	228,699 (65.44%)	
Previous		192 (31.32%)	91,310 (26.13%)	
Current		91 (14.85%)	29,469 (8.42%)	
Daily numbers of cigarette smoking (cigs/d) ^#^				
Previous cigarette smoking (cigs/d) ^#^	(522/320,009)	7.84	5.42	5.43
Current cigarette smoking (cigs/d) ^#^	(420/258,332)	3.85	1.75	1.75
Alcohol intake frequency	(725/411,611)			0.002
Never		67 (9.24%)	33,338 (8.10%)	
Ocassions,1–3 times/m		149 (20.55%)	93,334 (22.68%)	
1–4 times/w		326 (44.97%)	202,312 (49.15%)	
Daily		183 (25.24%)	82,627 (20.07%)	
BMI	(728/412,922)			0.001
Normal or Underweight (BMI < 25)		191 (26.24%)	135,077 (32.71%)	
Overweight (25 ≤ BMI < 30)		324 (44.51%)	174,640 (42.29%)	
Obese (BMI ≥ 30)		213 (29.26%)	103,205 (24.99%)	
Waist Circumference (cm) ^#^	(726/411,101)	93.77	90.38	<0.0001
Waist Hip Ratio (WHR) (Continuous) ^#^	(726/411,027)	0.90	0.87	0.001
Waist Hip Ratio (WHR) (Category)	(726/411,027)			<0.0001
Normal (M: <0.90, F: <0.85)		278 (38.29%)	208,189 (50.65%)	
Abdominal obesity (M: ≥0.90, F: ≥0.85)		448 (61.71%)	202,838 (49.35%)	
Processed meat consuming frequency	(726/410,971)			0.515
Never		57 (7.85%)	38,624 (9.40%)	
<1 time/week		221 (30.08%)	123,619 (30.08%)	
1 time/week		202 (27.82%)	119,505 (29.08%)	
2–4 times/week		218 (30.035%)	112,464 (27.37%)	
5–6 times/week		22 (3.03%)	13,280 (3.23%)	
≥1 time/day		6 (0.83%)	3479 (100%)	
Medical history-related variables				
Pancreatitis	(728/412,922)			<0.0001
No		719 (98.76%)	412,044 (99.79)	
Yes		9 (1.24%)	878 (0.21%)	
Diabetes Mellitus	(728/412,922)			<0.0001
No		648 (89.01%)	391,690 (94.86%)	
Yes		80 (10.99%)	21,232 (5.14%)	
Hepatitis B	(728/412,922)			0.645
No		728 (100%)	412,802 (99.97%)	
Yes		0 (0%)	120 (0.03%)	
Cholecystitis	(728/412,922)			0.759
No		727 (99.86%)	412,504 (99.90%)	
Yes		1 (0.14%)	418 (0.1%)	
Helicobacter Pylori Infection	(728/412,922)			0.896
No		726 (99.73%)	411,678 (99.70%)	
Yes		2 (0.27%)	1244 (0.30%)	
Systemic Lupus Erythematosis (SLE)	(728/412,922)			0.341
No		728 (100%)	412,409 (99.88%)	
Yes		0 (0%)	513 (0.12%)	

^#^ mean-value, * Chi-square test statistic or Student *t*-test for mean values.

**Table 2 cancers-14-04991-t002:** Pancreatic cancer (PaCa) relative risk (RR) of non-modifiable and modifiable factors.

Characteristic Variables	No. (PaCa Cases/Non-PaCa Controls)	RR	95% CI	*p*-Value
Non-modifiable factors				
Gender *	(728/412,922)			
Female		Ref.		
Male		1.27	(1.10–1.47)	0.001
Age (Continuous) **	(728/412,922)	1.03	(1.02–1.04)	<0.0001
Ethnic group	(725/410,536)			
White		Ref.		
Mix		0.70	(0.23–2.18)	0.541
Asian		0.62	(0.33–1.16)	0.137
Black		0.49	(0.22–1.10)	0.083
Chinese		1.27	(0.41–3.94)	0.682
Other		0.58	(0.22–1.56)	0.281
Modifiable factors: lifestyle-related factors				
Cigarette smoking status	(613/349,478)			
Never		Ref.		
Previous		1.36	(1.13–1.62)	0.001
Current		2.16	(1.71–2.73)	<0.0001
Alcohol intake frequency	(725/411,611)			
Never		Ref.		
Ocassions, 1–3 times/m		0.81	(0.61–1.08)	0.152
1–4 times/w		0.79	(0.61–1.03)	0.080
Daily		1.03	(0.78–1.37)	0.813
BMI	(728/412,922)			
Normal or underweight (BMI < 25)		Ref.		
Overweight (25 ≤ BMI < 30)		1.23	(1.03–1.48)	0.023
Obese (BMI ≥ 30)		1.39	(1.14–1.69)	0.001
Waist Circumference (cm)	(726/411,101)	1.02	(1.01–1.02)	<0.0001
Waist Hip Ratio (WHR) (Category)	(726/411,027)			
Normal (M: <0.90, F: <0.85)		Ref.		
Abdominal obesity (M: ≥0.90, F: ≥0.85)		1.56	(1.32–1.84)	<0.0001
Processed meat consuming	(726/410,971)			
Never		Ref.		
<1 time/week		1.17	(0.88–1.57)	0.281
1 time/week		1.08	(0.80–1.45)	0.627
2–4 times/week		1.20	(0.89–1.62)	0.224
5–6 times/week		1.01	(0.62–1.67)	0.959
≥1 time/day		1.06	(0.46–2.46)	0.894
Modifiable factors: medical history-related factors				
Pancreatitis	(728/412,922)			
No		Ref.		
Yes		5.54	(2.88–10.65)	<0.0001
Diabetes Mellitus	(728/412,922)			
No		Ref.		
Yes		2.08	(1.64–2.63)	<0.0001
Cholecystitis	(728/412,922)			
No		Ref.		
Yes		1.37	(0.19–9.73)	0.752
Helicobacter Pylori Infection	(728/412,922)			
No		Ref.		
Yes		0.89	(0.22–3.56)	0.870

All adjusted for age and gender; * Adjusted for age only; ** Adjusted for gender only.

**Table 3 cancers-14-04991-t003:** Pancreatic cancer (PaCa) relative risk (RR) of different daily cigarette smoking numbers. among former and current smokers.

Characteristic Variables	No. (PaCa Cases/Non-PaCa Controls)	RR	95% CI	*p*-Value
Daily numbers of previous cigarette smoking (cigs/d) (Continuous)	(522/320,009)	1.02	(1.01–1.02)	0.0001
Daily numbers of previous cigarette smoking (cigs/d) (Categorical)				
0		Ref.		
1–10		1.09	(0.79–1.53)	0.594
11–15		0.83	(0.53–1.30)	0.405
16–20		1.40	(1.08–1.82)	0.011
≥21		2.03	(1.54–2.68)	0.0001
Daily numbers of current cigarette smoking (cigs/d) (Continuous)		1.04	(1.03–1.05)	0.0001
Daily numbers of current cigarette smoking (cigs/d) (Categorical)				
0		Ref.		
1–10		1.30	(0.82–2.06)	0.268
11–15		2.34	(1.54–3.58)	0.0001
16–20		2.61	(1.77–3.87)	0.0001
≥21		3.46	(2.24–5.35)	0.0001

**Table 4 cancers-14-04991-t004:** Population attributable fraction (PAF) of modifiable risk factors among population and sub-population.

Characteristic Variables	PAF in Population	95% CI	PAF in Subpopulation	95% CI
Cigarette smoking status				
Never	Ref.		Ref.	
Previous	0.10	(0.03–0.15)	0.26	(0.11–0.38)
Current	0.16	(0.10–0.22)	0.54	(0.42–0.63)
BMI				
Normal or underweight (BMI < 25)	Ref.		Ref.	
Overweight (25 ≤ BMI < 30)	0.11	(0.01–0.21)	0.19	(0.03–0.32)
Obese (BMI ≥ 30)	0.17	(0.06–0.26)	0.28	(0.12–0.41)
Waist Hip Ratio (WHR) (Category)				
Normal (M: <0.90, F: <0.85)	Ref.		Ref.	
Abdominal obesity (M: ≥0.90, F: ≥0.85)	0.22	(0.14–0.30)	0.36	(0.24–0.46)
Pancreatitis				
No	Ref.		Ref.	
Yes	0.01	(0.002–0.02)	0.82	(0.65–0.91)
Diabetes Mellitus				
No	Ref.		Ref.	
Yes	0.06	(0.03–0.08)	0.52	(0.39–0.62)

## Data Availability

Data can only be accessed from the UK Biobank via the approved application. The data is owned by the UK Biobank (www.ukbiobank.ac.uk (accessed on 22 August 2022)) and as researchers we are not entitled to republish or otherwise make available any UK Biobank data at the individual participant level. The UK Biobank, however, is open to all bona fide researchers anywhere in the world. Detailed access procedures can be found by following this link: https://www.ukbiobank.ac.uk/media/omtl1ie4/access-procedures-2011-1.pdf (accessed on 22 August 2022). The data used in this study (application number 5974) can be requested by applying through the UK Biobank Access Management System (www.ukbiobank.ac.uk/register-apply (accessed on 22 August 2022)).

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
