# Peer review of "Risk Factors Associated with Pancreatic Cancer in the UK Biobank Cohort"

_cancers, 2022, doi:10.3390/cancers14204991_

Round 1

Reviewer 1 Report

Dear Authors,

The article entitled: Risk Factors associated with Pancreatic Cancer in the UK Biobank Cohort, by Te-Min Ke et al. provides a large population study and could serve as a prevention study. Moreover, it is overall well written; however, it has several flaws that has to be amended before consider publication. Please find below my points:

Minors

1.-Pancreatitis history is a non-modifiable factor.

Major points

1.- The introduction concerning risk factor not only modifiable but also non-modifiable factors is very limited. I strongly suggest to include more details about risk factors.

2.-To be a research article other important non-modifiable risk factors have been ignored. For example: familial cancer syndrome (non-modifiable).

3.-Concerning Diabetes Mellitus, it is crucial to know what kind of treatment the patients are following. Is not the same at molecular level the Metformine-based treatment than other drugs. I would include metformine or other treatments in the analyses.

4.-Since very limited variables have been taken into consideration in this research article I strongly recommend to include other analyses involving patient survival and chemotherapy response. Then, Kaplan-Meier curves for overall survival and progression-free survival and adjuvant chemotherapy response according to these risk factors would be appreciated and very interesting for pancreatic cancer research community.

5.-I miss to know the stage of the pancreatic cancer patients included in the study.

Author Response

Response to Reviewer 1

We would like to thank reviewer 2 for his/her comments. We have addressed all raised points as per below.

Minors

Point 1: Pancreatitis history is a non-modifiable factor.

Response 1: Please provide your response for Point 1. (in red)

We agree that pancreatic history is a non-modifiable factor. However, in this study, we defined modifiable factors as the exposures that occurred before pancreatic cancer (PaCa) diagnosis and therefore could potentially be classified as preventable or modifiable. Similarly, non-modifiable factors were the exposure factors that were present before the PaCa occurred; however, they could not be prevented or modifiable. In a literature review[1]: Modifiable and Non-Modifiable Risk Factors for the Development of Non-Hereditary Pancreatic Cancer, both acute and chronic pancreatitis were also categorized as the modifiable factor. We added the definition of modifiable and non-modifiable risk factors in the Methodology 2.3 exposures to clarify our definition of these terms.

Modifiable factors were defined as the exposures that occurred before PaCa diagnosis and could be preventable or modifiable. Non-modifiable factors were identified as the exposure factors that were presented before the PaCa diagnosis and could not be prevented or modified.” 

Major

Point 1: The introduction concerning risk factor not only modifiable but also non-modifiable factors is very limited. I strongly suggest to include more details about risk factors.

Response 1: Please provide your response for Point 2. (in red)

We thank you reviewer 1 for the comment which we agree with. We added further details to the introduction section.

Previously published literature[2-4] reported many potential PaCa risk factors. Some non-modifiable risk factors showed probable evidence of PaCa, including ageing, male gender, African American ethnicity, non-O blood type, family history, inherited syndromes including Peutz-Jeghers syndrome, Hereditary pancreatitis and Lynch syndrome, and germline mutation (CDKN2A, TP53, MLH1, ATM, BRCA2, MSH2, MSH6, PALB2, BRCA1). Other modifiable risk factors with probable or convincing evidence are cigarette smoking, heavy alcohol consumption, increased Body Mass Index (BMI) and abdominal obesity[5,6], chronic pancreatitis, Diabetic Mellitus (DM), hepatitis B, cholecystectomy[7,8], and periodontal disease. Nevertheless, there were still other ambiguous risk factors with inconclusive evidence, such as increased consumption of processed meat, Helicobacter Pylori infection, and Systemic Lupus Erythematosus (SLE) history[9,10].”

Point 2: To be a research article other important non-modifiable risk factors have been ignored. For example: familial cancer syndrome (non-modifiable).

Response 2:

In this UK Biobank cohort, we did not discuss the relative risk of PaCa for family history and inherited syndromes. This was due to the limitation of the availability of such data in the UK Biobank study. The focus of this paper was on epidemiological factors for pancreatic cancer. In terms of the genetic factors, we are currently preparing a further manuscript which focuses on the genetic factors and their interactions with the epidemiological factors. We have added the following text to the introduction.

Added text to the Discussion section:

Other familial-related factors such as non-O blood type, a kindred PaCa history, several familiar syndromes and gerlime mutation were reported to associate with higher PaCa risk[2-4]. However, some of the information is not available to obtain from the UK biobank. Furthermore, the focus of this paper was on epidemiological factors for pancreatic cancer.

Point 3: Concerning Diabetes Mellitus, it is crucial to know what kind of treatment the patients are following. Is not the same at molecular level the Metformine-based treatment than other drugs. I would include metformine or other treatments in the analyses.

Response 3:

Thank you for your suggestion. We did not aim to discuss the relationship between different anti-diabetic medications and PaCa risk. We aim to discover the relative risk and population attribution fraction of DM in the UK population. We have added the following text to the discussion.

Our results suggested that preventing DM could avoid more PaCa incidence cases. In terms of anti-diabetic medications, although there is some evidence showing that these medications are linked to the PaCa risk, however, the evidence is inconclusive. For example, some studies supported that taking insulin and sulfonylureas (SUs) may increase PaCa risk[11,12]. In contrast, metformin has been hypothesized to have a protective effect on PaCa risk[13,14]. A systematic review[15] showed no relationship between using metformin (OR=0.76, 95% CI: 0.57-1.03), insulin (OR=1.59, 95%CI: 0.85-2.96) and thiazolidinediones (TZDs) (OR=1.02, 95% CI: 0.81-1.30) and risk of developing PaCa. On the other hand, patients using SUs had a higher OR with 1.70 (95 %CI: 1.27-2.28). In contrast, a meta-analysis study[13] did not find a significant relationship between using SUs and PaCa risk. Furthermore, the interpretation of these results needs to consider reverse causality and protopathic bias[16]. Therefore, it still needs more prospective observational studies to evaluate the impact of anti-diabetic medications on PaCa risk. In our study, we did not investigate the effects of these anti-diabetic medications due to data availability.”

Point 4: Since very limited variables have been taken into consideration in this research article I strongly recommend to include other analyses involving patient survival and chemotherapy response. Then, Kaplan-Meier curves for overall survival and progression-free survival and adjuvant chemotherapy response according to these risk factors would be appreciated and very interesting for pancreatic cancer research community.

Response 4:
We appreciate your suggestions; it will be exciting to discover the overall survival and progression-free survival of the neoadjuvant, adjuvant chemo-radiotherapy, surgery, etc. However, our review aims to explore the PaCa risk factors in the UK population, and we anticipate applying our results in future prevention work and predictive model establishment. Furthermore, obtaining the details of treatment information in this UK Biobank study is limited. Therefore, using the UK Biobank cohort to discuss the prognosis of different treatments is not viable.  

Point 5: I miss to know the stage of the pancreatic cancer patients included in the study.      

Response 5:

In this study, the diagnosis of pancreatic cancer was recorded by ICD9 and 10. However, the UK Biobank did not have information on cancer stage information. This would be very informative to explore the different magnitudes of risk in different pathology.

We have added further reference to our modified manuscript.

  1. Olakowski, M.; Bułdak, Ł. Modifiable and Non-Modifiable Risk Factors for the Development of Non-Hereditary Pancreatic Cancer. Medicina (Kaunas) 2022, 58, doi:10.3390/medicina58080978.
  2. Klein, A.P. Pancreatic cancer epidemiology: understanding the role of lifestyle and inherited risk factors. Nat Rev Gastroenterol Hepatol 2021, 18, 493-502, doi:10.1038/s41575-021-00457-x.
  3. Hu, J.X.; Zhao, C.F.; Chen, W.B.; Liu, Q.C.; Li, Q.W.; Lin, Y.Y.; Gao, F. Pancreatic cancer: A review of epidemiology, trend, and risk factors. World journal of gastroenterology 2021, 27, 4298-4321, doi:10.3748/wjg.v27.i27.4298.
  4. Midha, S.; Chawla, S.; Garg, P.K. Modifiable and non-modifiable risk factors for pancreatic cancer: A review. Cancer Lett 2016, 381, 269-277, doi:10.1016/j.canlet.2016.07.022.
  5. Aune, D.; Greenwood, D.C.; Chan, D.S.; Vieira, R.; Vieira, A.R.; Navarro Rosenblatt, D.A.; Cade, J.E.; Burley, V.J.; Norat, T. Body mass index, abdominal fatness and pancreatic cancer risk: a systematic review and non-linear dose-response meta-analysis of prospective studies. Ann Oncol 2012, 23, 843-852, doi:10.1093/annonc/mdr398.
  6. Genkinger, J.M.; Spiegelman, D.; Anderson, K.E.; Bernstein, L.; van den Brandt, P.A.; Calle, E.E.; English, D.R.; Folsom, A.R.; Freudenheim, J.L.; Fuchs, C.S.; et al. A pooled analysis of 14 cohort studies of anthropometric factors and pancreatic cancer risk. Int J Cancer 2011, 129, 1708-1717, doi:10.1002/ijc.25794.
  7. Lin, G.; Zeng, Z.; Wang, X.; Wu, Z.; Wang, J.; Wang, C.; Sun, Q.; Chen, Y.; Quan, H. Cholecystectomy and risk of pancreatic cancer: a meta-analysis of observational studies. Cancer Causes Control 2012, 23, 59-67, doi:10.1007/s10552-011-9856-y.
  8. Uldall Torp, N.M.; Kristensen, S.B.; Mortensen, F.V.; Kirkegård, J. Cholecystitis and risk of pancreatic, liver, and biliary tract cancer in patients undergoing cholecystectomy. HPB (Oxford) 2020, 22, 1258-1264, doi:10.1016/j.hpb.2019.11.012.
  9. Seo, M.S.; Yeo, J.; Hwang, I.C.; Shim, J.Y. Risk of pancreatic cancer in patients with systemic lupus erythematosus: a meta-analysis. Clin Rheumatol 2019, 38, 3109-3116, doi:10.1007/s10067-019-04660-9.
  10. Song, L.; Wang, Y.; Zhang, J.; Song, N.; Xu, X.; Lu, Y. The risks of cancer development in systemic lupus erythematosus (SLE) patients: a systematic review and meta-analysis. Arthritis Res Ther 2018, 20, 270, doi:10.1186/s13075-018-1760-3.
  11. Lu, Y.; García Rodríguez, L.A.; Malgerud, L.; González-Pérez, A.; Martín-Pérez, M.; Lagergren, J.; Bexelius, T.S. New-onset type 2 diabetes, elevated HbA1c, anti-diabetic medications, and risk of pancreatic cancer. Br J Cancer 2015, 113, 1607-1614, doi:10.1038/bjc.2015.353.
  12. Bosetti, C.; Rosato, V.; Li, D.; Silverman, D.; Petersen, G.M.; Bracci, P.M.; Neale, R.E.; Muscat, J.; Anderson, K.; Gallinger, S.; et al. Diabetes, antidiabetic medications, and pancreatic cancer risk: an analysis from the International Pancreatic Cancer Case-Control Consortium. Ann Oncol 2014, 25, 2065-2072, doi:10.1093/annonc/mdu276.
  13. Soranna, D.; Scotti, L.; Zambon, A.; Bosetti, C.; Grassi, G.; Catapano, A.; La Vecchia, C.; Mancia, G.; Corrao, G. Cancer risk associated with use of metformin and sulfonylurea in type 2 diabetes: a meta-analysis. Oncologist 2012, 17, 813-822, doi:10.1634/theoncologist.2011-0462.
  14. Bodmer, M.; Becker, C.; Meier, C.; Jick, S.S.; Meier, C.R. Use of antidiabetic agents and the risk of pancreatic cancer: a case-control analysis. Am J Gastroenterol 2012, 107, 620-626, doi:10.1038/ajg.2011.483.
  15. Singh, S.; Singh, P.P.; Singh, A.G.; Murad, M.H.; McWilliams, R.R.; Chari, S.T. Anti-diabetic medications and risk of pancreatic cancer in patients with diabetes mellitus: a systematic review and meta-analysis. Am J Gastroenterol 2013, 108, 510-519; quiz 520, doi:10.1038/ajg.2013.7.
  16. Sharma, A.; Chari, S.T. Pancreatic Cancer and Diabetes Mellitus. Current Treatment Options in Gastroenterology 2018, 16, 466-478, doi:10.1007/s11938-018-0197-8.

Reviewer 2 Report

This population-based cohort study (UK Biobank) investigated the population attributable fraction (PAF) of non-modificable und modificable risk factors for pancreatic cancer (PaCa).

The UK-national-based dataset contained a substantial number of 502.413 participants from years 2006 to 2010 with a median follow-up time up to 8.2 years. 88.6762 participants were excluded because of concurrent other neoplasms and 91 participants with prevalent pancreatic cancer.

The introduction, methods/materials section, results and discussion part are written well and understandable.

The explorative data analysis is designed properly and the presentation of results is clear and comprehensible. Non-modificable risk factors for PaCa (age, gender, ethnicity), modificable lifestyle-related risk factors (smoking,  overweight, abdominal obesity, BMI, waist circumstance, etc.) as well as modificable medical history-related risk factors (diabetes, pancreatitis) were evaluated in the context of PaCa.

The results of this cohort study match the published data in several recent reports (Pancreatic cancer epidemiology: understanding the role of lifestyle and inherited risk factors, Alison P. Klein, Nat Rev Gastroenterol Hepatol. 2021 July ; 18(7): 493–502; Epidemiology of pancreatic cancer, Milena Ilic, Irena Ilic, World J Gastroenterol 2016 November 28; 22(44): 9694-9705; Pancreatic cancer: A review of clinical diagnosis, epidemiology, treatment and outcomes, Andrew McGuigan, Paul Kelly, Richard C Turkington,  et al., World J Gastroenterol 2018 November 21; 24(43): 4846-4861)

Major remark 1:

Unfortunately, familial history of panreatic cancer, hereditary pancreatitis and potential role of involvement of inherited predisposition genes for PaCa / genetic susceptibility  (i.e. BRCA, MSI ...) were not considered in the part of non-modificable risk factors. The factors blood group and intestinal microbioma also could be mentioned.

The authors should comment on this in the discussion part (may be some of these data are not available the UK Biobank).

Major remark 2:

The authors conclude, that this report should aim to heighten the public awareness for modificable risk factors for PACa and the identified risk factors should be taken into account in (inter)national PACa prevention programs in oder to decrease the incidence of PaCa.

 This UK dataset - with a vast majority containing caucasian / european participants - should be discussed against the background of worldwide populations considering different ethnicities and consequential different populations based-risks.

Author Response

Response to Reviewer 2  

We would like to thank reviewer 2 for his/her comments. We have addressed all raised points as per below.

Major comment

Point 1: Unfortunately, familial history of panreatic cancer, hereditary pancreatitis and potential role of involvement of inherited predisposition genes for PaCa / genetic susceptibility (i.e. BRCA, MSI ...) were not considered in the part of non-modificable risk factors. The factors blood group and intestinal microbioma also could be mentioned.

The authors should comment on this in the discussion part (may be some of these data are not available the UK Biobank).

Response 1:

We appreciated the reviewer’s suggestion, and we have revised our manuscript by adding other potential risk factors in the introduction section. The focus of this paper was on epidemiological factors for pancreatic cancer. In terms of the genetic factors, we are currently preparing a further manuscript which focuses on the genetic factors and their interactions with the epidemiological factors.

The newly added texts in the introduction section are as shown below.

Previously published literature[1-3] reported many potential PaCa risk factors. Some non-modifiable risk factors showed probable evidence of PaCa, including ageing, male gender, African American ethnicity, non-O blood type, family history, inherited syndromes including Peutz-Jeghers syndrome, Hereditary pancreatitis and Lynch syndrome, and germline mutation (CDKN2A, TP53, MLH1, ATM, BRCA2, MSH2, MSH6, PALB2, BRCA1). Other modifiable risk factors with probable or convincing evidence are cigarette smoking, heavy alcohol consumption, increased Body Mass Index (BMI) and abdominal obesity[4,5], chronic pancreatitis, Diabetic Mellitus (DM), hepatitis B, cholecystectomy[6,7], and periodontal disease. Nevertheless, there were still other ambiguous risk factors with inconclusive evidence, such as increased consumption of processed meat, Helicobacter Pylori infection, and Systemic Lupus Erythematosus (SLE) history[8,9].”

We also added the explanation for omitting discussion of some other potential risk factors to the discussion section as per below.

 “Other familial-related factors such as non-O blood type, a kindred PaCa history, several familiar syndromes and gerlime mutation were reported to associate with higher PaCa risk[1-3]. However, some of the information is not available to obtain from the UK biobank. Furthermore, the focus of this paper was on epidemiological factors for pancreatic cancer. ”

“4.2. Modifiable medical history-related risk factors:

For other potential medical-related risk factors, periodontal disease was associated with a higher risk of PaCa[1,2], which may be related to the alterations of the oral microbiome; however, the mechanism has not been well established. Nevertheless, we were unable to obtain this information from the UK Biobank.”

Point 2: The authors conclude, that this report should aim to heighten the public awareness for modificable risk factors for PACa and the identified risk factors should be taken into account in (inter)national PACa prevention programs in oder to decrease the incidence of PaCa.

 This UK dataset - with a vast majority containing caucasian / european participants - should be discussed against the background of worldwide populations considering different ethnicities and consequential different populations based-risks.

Response 2: 

We agree with the reviewer. We added this issue in 4.3. Strengths and Limitations:

“Finally, most of the participants were enrolled from the Caucasian population. Therefore, our conclusion should be extrapolated with caution to non-Caucasian populations. “

We also modified our reference list by adding the list below to the main manuscript.

Reference:

  1. Klein, A.P. Pancreatic cancer epidemiology: understanding the role of lifestyle and inherited risk factors. Nat Rev Gastroenterol Hepatol 2021, 18, 493-502, doi:10.1038/s41575-021-00457-x.
  2. Hu, J.X.; Zhao, C.F.; Chen, W.B.; Liu, Q.C.; Li, Q.W.; Lin, Y.Y.; Gao, F. Pancreatic cancer: A review of epidemiology, trend, and risk factors. World journal of gastroenterology 2021, 27, 4298-4321, doi:10.3748/wjg.v27.i27.4298.
  3. Midha, S.; Chawla, S.; Garg, P.K. Modifiable and non-modifiable risk factors for pancreatic cancer: A review. Cancer Lett 2016, 381, 269-277, doi:10.1016/j.canlet.2016.07.022.
  4. Aune, D.; Greenwood, D.C.; Chan, D.S.; Vieira, R.; Vieira, A.R.; Navarro Rosenblatt, D.A.; Cade, J.E.; Burley, V.J.; Norat, T. Body mass index, abdominal fatness and pancreatic cancer risk: a systematic review and non-linear dose-response meta-analysis of prospective studies. Ann Oncol 2012, 23, 843-852, doi:10.1093/annonc/mdr398.
  5. Genkinger, J.M.; Spiegelman, D.; Anderson, K.E.; Bernstein, L.; van den Brandt, P.A.; Calle, E.E.; English, D.R.; Folsom, A.R.; Freudenheim, J.L.; Fuchs, C.S.; et al. A pooled analysis of 14 cohort studies of anthropometric factors and pancreatic cancer risk. Int J Cancer 2011, 129, 1708-1717, doi:10.1002/ijc.25794.
  6. Lin, G.; Zeng, Z.; Wang, X.; Wu, Z.; Wang, J.; Wang, C.; Sun, Q.; Chen, Y.; Quan, H. Cholecystectomy and risk of pancreatic cancer: a meta-analysis of observational studies. Cancer Causes Control 2012, 23, 59-67, doi:10.1007/s10552-011-9856-y.
  7. Uldall Torp, N.M.; Kristensen, S.B.; Mortensen, F.V.; Kirkegård, J. Cholecystitis and risk of pancreatic, liver, and biliary tract cancer in patients undergoing cholecystectomy. HPB (Oxford) 2020, 22, 1258-1264, doi:10.1016/j.hpb.2019.11.012.
  8. Seo, M.S.; Yeo, J.; Hwang, I.C.; Shim, J.Y. Risk of pancreatic cancer in patients with systemic lupus erythematosus: a meta-analysis. Clin Rheumatol 2019, 38, 3109-3116, doi:10.1007/s10067-019-04660-9.
  9. Song, L.; Wang, Y.; Zhang, J.; Song, N.; Xu, X.; Lu, Y. The risks of cancer development in systemic lupus erythematosus (SLE) patients: a systematic review and meta-analysis. Arthritis Res Ther 2018, 20, 270, doi:10.1186/s13075-018-1760-3.

Round 2

Reviewer 1 Report

Dear authors, 

Thanks so much for the responses to my comments. Please find below my replies:

1) modifiable and non-modifiable factor are what they are and consensual internationally. Therefore, I will not accept any personal consideration of modifiable and non-modifiable factor. Please change the manuscript accordingly.

2)To suffer from a pancreatic cancer with stage I differs from suffer from a stage IV at diagnosis. To have this information and evaluate risk factors according to stage I or metastasic disease is crucial for this research  

Reviewer 2 Report

Point by point reply is fine.